# Effect of goal-directed mobilisation intervention compared with standard care on physical activity among medical inpatients: protocol for the GoMob-in randomised controlled trial

Fabian D Liechti ⬡ ,[1] Jeannelle Heinzmann,[1] Joachim M Schmidt Leuenberger,[2] Andreas Limacher,[3] Maria M Wertli ⬡ ,[1,4] Martin L Verra[2]

¹Department of General Internal Medicine, Inselspital, Bern University Hospital, University of Bern, Bern, Switzerland
²Department of Physiotherapy, Inselspital, Bern University Hospital, Bern, Switzerland
³CTU Bern, University of Bern, Bern, Switzerland
⁴Department of General Internal Medicine, Kantonsspital Baden, Baden, Switzerland

**Correspondence to**
Dr Fabian D Liechti;
fabian.liechti@med.unibe.ch

## ABSTRACT

**Introduction** Despite the fact that immobilisation is a major contributor to morbidity and mortality, patients hospitalised in general internal medicine (GIM) wards spend up to 50% of time in bed. Previous studies in selected patient populations showed increased mobility after implementation of goal-directed mobilisation (GDM). Due to the study design used so far, the degree of evidence is generally low. The effect of GDM on clinical outcomes and economically relevant indicators in patients hospitalised in GIM wards is currently unknown. This study aims to evaluate a GDM intervention compared to standard care on physical activity (de Morton Mobility Index, DEMMI) in medical inpatients.

**Methods and analysis** GoMob-in is a randomised, controlled, open-label study with blinded outcome assessment. We plan to enrol 160 inpatients with indication for physiotherapy on GIM wards of a tertiary hospital in Bern, Switzerland. Adult patients newly hospitalised on GIM wards will be included in the study. The primary outcome will be the change in the DEMMI score between baseline and 5 days. Secondary outcomes are change of DEMMI (inclusion to hospital discharge), mobilisation time (inclusion to day 5, inclusion to discharge), in-hospital delirium episodes, number of in-hospital falls, length of stay, number of falls within 3 months, number of re-hospitalisations and all-cause mortality within 3 months, change in independence during activities of daily living, concerns of falling, and quality of life within 3 months and destination after 3 months. Patients in the intervention group will be attributed a regularly updated individual mobility goal level made visible for all stakeholders and get a short educational intervention on GDM.

**Ethics and dissemination** This study has been approved by the responsible Ethics Board (Ethikkommission Bern/2020–02305). Written informed consent will be obtained from participants before study inclusion. Results will be published in open access policy peer-reviewed journals.

**Trial registration number** NCT04760392.

## INTRODUCTION

Decreased mobility during hospitalisation is associated with muscle loss, risk of falls,

### Strengths and limitations of this study

⇒ This is the first study evaluating goal-directed mobilisation in hospitalised general internal medicine patients using blinded outcome assessment.
⇒ The primary endpoint physical activity will be assessed by change in the de Morton Mobility Index score between baseline and day 5, a reliable tool without floor or ceiling effects.
⇒ Activity will be recorded by an accelerometer to record the effect of the intervention.
⇒ The study will also assess independence during activities of daily living, quality of life, number of re-hospitalisations and all-cause mortality at 3 months follow-up.
⇒ The study design is limited by the fact that blinding is not feasible for the interventions and some outcomes.

delirium, functional decline and an increased mortality, especially in the elderly.[1–4] Bedrest of 10 days in healthy elderly resulted in a substantial loss of muscle strength, power, aerobic capacity and a reduction in physical activity.[5] Evidence suggests that interventions that promote early mobility have the potential to decrease the risk of falls,[6 7] to reduce functional decline during hospitalisation,[8] to shorten duration of delirium[9] and to reduce the median length of hospital stay (LOS) and mortality.[9 10] Therefore, early mobilisation is recommended in most patients to prevent functional decline and frailty.[11 12] Despite recommendations to mobilise patients early, a low level of physical activity is still common and many patients spend most of the time in bed during hospitalisation, irrespective of patient age.[3 12–14] In a recent Dutch study, inpatients spent almost half of the time in bed and were active only 10% of the time.[15] Most

**BMJ**

studies assessing interventions on mobilisation focused on selected patients such as older adults or those admitted to intensive care units. Further, most studies failed to monitor mobility or long-term functional outcomes or had a retrospective design.[2 12 16–18]

Implementation of early mobilisation in general internal medicine (GIM) remains challenging and staff resources-demanding, because mobilisation is usually supervised by healthcare professionals such as physiotherapists. Individual goal-achievement strategies have the potential to complement physiotherapy in rehabilitation.[12 19] Thereby, a mobility goal is generally defined depending on the individual potential of the patient and communicated to the interprofessional team caring for the patient. Goal-directed mobility (GDM) programmes in GIM showed in temporal cohorts a reduction in LOS and a decrease of falls.[20 21] Most evidence stems from studies which usually integrated several other organisational changes or were non-randomised trials, for example, implementation of a mobility scale and corresponding mobility goal in hospital wards.[22] However, pre-post settings do not consider general trends, for example, reduction in LOS and comparators may differ significantly. In one randomised controlled trial, a mobility programme using goal-setting reduced the decline in activities of daily living during acute hospitalisation.[7] However, it included only a small number of older veterans was labour intense and lacked long-term follow-up.[7]

Due to the low number of studies of limited methodological quality, the level of evidence is low and the impact of GDM on clinical and economic outcomes in GIM patients is widely unknown.[23]

Therefore, this study aims to evaluate a standardised GDM programme in acutely hospitalised medical inpatients on functional capacity during hospitalisation and up to 3 months after study inclusion. All patients will receive physiotherapy treatment as indicated by the physician in charge. Compared with a control group with standard care, patients in the intervention group will receive additional instructions for GDM, physiotherapists will work with goal-setting and the goal will be indicated for all stakeholders at the bedside.

## METHODS AND ANALYSIS

The methods reporting of this randomised trial will follow the recommendations of the SPIRIT statement (table 1).[24]

## Study setting

This is an investigator initiated trial. The funder has no role in the study design, the trial oversight, data collection, analysis of the study and publication of the results. The Goal-directed Mobilization of Medical Inpatients (GoMob-in) trial is a pragmatic, randomised, controlled, open-label study with a 3-month follow-up and a blind outcome assessment, being conducted at the Department

**Table 1** Trial registration data

| Data category | Information |
|---|---|
| Primary registry and trial identifying number | ClinicalTrials.gov NCT04760392 |
| Date of registration in primary registry | 18 February 2021 |
| Secondary numbers | SNCTP000004280, BASEC2020-02305 |
| Source(s) of monetary or material support | Swiss Society of General Internal Medicine (SGAIM) Foundation, Bern, Switzerland Fondation Sana, Bern, Switzerland |
| Primary sponsor | Swiss Society of General Internal Medicine (SGAIM) Foundation, Bern, Switzerland (grant number: not applicable) |
| Secondary sponsor | Fondation Sana, Bern, Switzerland (grant number GF 2021–0048) |
| Contact for public queries | FDL fabian.liechti@insel.ch |
| Contact for scientific queries | FDL Department of General Internal Medicine, Inselspital, Bern University Hospital, Bern, Switzerland |
| Public title | Goal-directed mobilisation of medical inpatients—a randomised, controlled trial (GoMob-in trial) |
| Scientific title | Goal-directed mobilisation of medical inpatients—a randomised, controlled trial (GoMob-in trial) |
| Study setting | Number of study centre(s): 1 Type of study centre(s): University Hospital, Department of General Internal Medicine Location, Country: Bern, Switzerland |
| Health condition(s) or problem(s) studied | Inpatient mobility |
| Intervention(s) | Intervention group: goal-directed mobilisation |
| | Control group: standard of care |
| Key inclusion and exclusion criteria | Ages eligible for study: 18 years or older Sexes eligible for study: both Accepts healthy volunteers: no |
| | Inclusion criteria: acute hospitalisation on the general internal medicine ward, indication for physiotherapy |
| | Exclusion criteria: inability to follow study procedures and give informed consent themselves, expected hospital stay for <5 days, bedrest, injuries or relevant neurological deficits one or both lower extremities directly impairing walking capacity, terminal illness |
| Study type | Interventional Allocation: randomised intervention Masking: single blind (primary outcome assessor) Primary purpose: prevention |
| Date of first enrolment | 14.09.2021 |
| Target sample size | 160 |
| Recruitment status | Recruiting |

Continued

**Table 1** Continued

| Data category | Information |
|---|---|
| Primary outcome | Change of the de Morton Mobility Index (DEMMI) score between baseline and day 5 |
| Key secondary outcome | 1. Change in DEMMI score between baseline and discharge.<br>2. Mobilisation time measured by accelerometer between inclusion and day 5.<br>3. Mobilisation time between inclusion and discharge.<br>4. Number of delirium episodes.<br>5. Number of in-hospital falls.<br>6. Length of hospital stay.<br>7. Total number of falls (with/without injuries) within 3 months.<br>8. Number of rehospitalisations and all-cause mortality within 3 months.<br>9. Independence during activities of daily living: change in Barthel index between study inclusion and 3 months.<br>10. Change in FES-I between baseline and 3 months.<br>11. Quality of life: change in EuroQol (EQ-5D-5L) between study inclusion and 3 months.<br>12. Destination at 3 months after study inclusion (may include: death, acute care hospital, rehabilitation, home, nursing home, others). |

DEMMI, de Morton Mobility Index; FES-I, Falls Efficacy Scale-International.

of General Internal Medicine, Inselspital, Bern University Hospital, Bern, Switzerland.

### Objectives

To evaluate the effect of GDM compared with standard care in hospitalised patients in GIM on physical activity as assessed by the de Morton Mobility Index (DEMMI) score.

### Hypothesis

GDM improves physical activity assessed by DEMMI compared with standard of care in hospitalised patients in GIM.

### Eligibility criteria

#### Inclusion criteria

Subjects fulfilling all of the following inclusion criteria are eligible for the study:

► Acute hospitalisation on the GIM ward (elective or urgent admission, at latest on the second day after internal or external hospital admission).
► Age 18 years or older.
► Indication for physiotherapy.
► Informed consent as documented by signature (written informed consent).

### Exclusion criteria

The presence of any one of the following exclusion criteria will lead to exclusion of the subjects:

► Inability to follow study procedures, that is, due to language problems (unable to read, speak or understand German), psychological disorders, severe dementia (defined as to levels 5–7 in the Global Deterioration Scale),[25] blindness, patients unable to provide informed consent themselves.
► Expected hospital stay for <5 days.
► Medically indicated bedrest for more than 24 hours, for example, after surgery.
► Injuries or relevant neurological deficits one or both lower extremities directly impairing walking capacity (eg, fractures, hemiplegia, previous use of a wheelchair or bedriddenness).
► Terminal illness (ie, end-of-life care, dying phase).
► Pregnancy or breast feeding.
► Previous enrolment in this study.
► A study participant in the same patient room.
► Enrolment of the investigator, his/her family members, employees and other dependent persons.

### Estimated sample size and power

The sample size calculation was based on the primary outcome of a clinically meaningful change in the DEMMI score between baseline and 5 days of follow-up. The DEMMI score ranges from 0 to 100 points with 0 indicating no physical activity and 100 indicating maximal physical activity. Based on previous studies on changes in DEMMI scores during short-term hospitalisation of roughly 10 days,[26 27] we expect a difference in change between the two groups of 5 score points. In an observational study of participants with low and high activity, SD for changes in the DEMMI score was 7.8 and 8.5 points, respectively (personal communication).[26] In a study on older adult inpatients, the SD was between 12.5 and 15.4 points for absolute DEMMI scores at admission and discharge, respectively.[27] The SD for changes is typically smaller than for absolute values; we therefore expect a rather conservative SD of about 10 points for the change in our study.

To detect a difference in change of 5 score points with a SD of 10 points, at a two-sided alpha level of 0.05 with a power of 0.8, a sample size of 2×64 participants is required. Based on an estimated drop-out rate of 20%, we aim to include 160 patients into the study (80 in each group).

### Primary outcome

The primary outcome will be the change in physical activity level measured by performance in the DEMMI score between baseline and 5 days of follow-up.[28] Trained study personnel will evaluate physical activity by using DEMMI score at baseline before randomisation, while a physiotherapist blinded to the group allocation will assess DEMMI score 5 days later.

## Secondary outcome

1. Change in DEMMI score between baseline at study inclusion and discharge. Patients hospitalised for more than 2 weeks on GIM wards will be assessed on day 14 after study inclusion instead of at hospital discharge.
2. Percentage of time moving between study inclusion and day 5. Activity will be measured using an accelerometer (GENEActiv, Activinsights, Kimbolton, Cambridgeshire, UK) validated to assess activity in the hospital setting.[13 14 29]
3. Percentage of mobilisation time between baseline and discharge or at 14 days (whichever occurs first).
4. Number of in-hospital delirium episodes (follow-up max. 14 days).
5. Number of in-hospital falls (follow-up max. 14 days)
6. LOS defined as days between hospital admission and discharge.
7. Total number of falls (with/without injuries) within 3 months after study inclusion.
8. Number of rehospitalisations and all-cause mortality within 3 months after study inclusion.
9. Change in independence during activities of daily living (Barthel index) between baseline and 3 months after study inclusion.[30 31]
10. Change in Falls Efficacy Scale-International (FES-I) between baseline and 3 months after study inclusion.[32 33]
11. Change in quality of life (EQ-5D-5L) (visual analogue scale (EQ-VAS) and mobility dimension) between baseline and 3 months after study inclusion.[34–36]
12. Discharge destination (death, acute care hospital, rehabilitation, home, nursing home).
13. Reaching the minimal clinically important difference (MCID) of a change in DEMMI score (9 points) between baseline and discharge.[28 37]

## Study procedure

This study will generate evidence about a specific, easily implementable intervention using little resources to promote mobility and evaluate its impact on significant clinical outcomes of GIM inpatients (figure 1). Study recruitment started on 13 September 2021. Study completion is expected by 15 December 2022.

### Prescreening and recruitment

All patients hospitalised on the GIM department will be prescreened during weekdays within the first 2 days of hospitalisation for study participation.

### Baseline assessment

Study personnel will obtain written informed consent (online supplemental file 1) by the participant. After study inclusion, each patient receives a structured systematic assessment by the study coordinator including (table 2):

► Sociodemographic and anthropometric data (eg, age, gender, weight and height, zip code (rural versus urban living area), marital status, nurse visit at home (German: Spitex), insurance coverage (private or

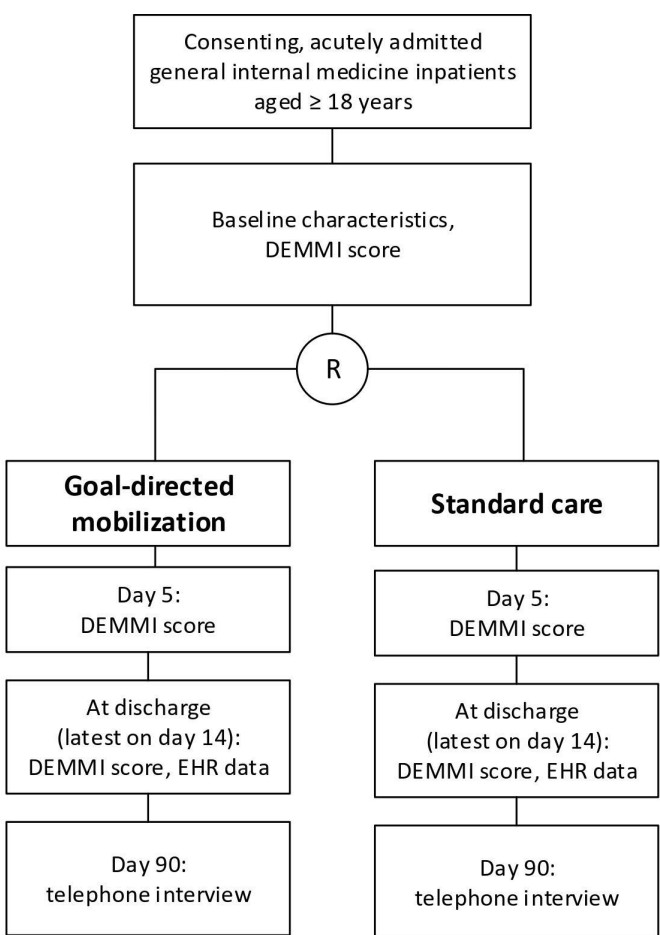

**Figure 1** Study flowchart. DEMMI, de Morton Mobility Index; EHR, electronic health record; R, randomisation.

not), admission mode (elective or urgent), transfer from intensive care unit, hospitalisation in the last 6 months), previous use of mobility aids.
► Comorbidities will be reported by the participant, using an established, systematic method as previously described.[38]
► Functional status using Barthel index. The Barthel index is one of the most widely used and reliable functional outcome measures and is measured through an established and validated questionnaire where higher scores on a scale between 0 and 100 indicate greater independence in activities of daily living.[30] It can be conducted by telephone and a validated German version is available.[31] If the patient is not able to answer the questions, the Barthel index will be administered by interviewing the ward nurse caring for the patient, a family member or any other person caring for the patient before hospitalisation (eg, home care nurse).
► Concerns of falling using FES-I. FES-I is a validated 16-item tool to evaluate concerns of falling, with 16 score points indicating no concerns of falling and 64 score points indicating high levels of concern.[32] It can be conducted by telephone and a validated German version is available.[33]

**Table 2** Summary of study procedures based on SPIRIT schedule

| | First visit | Second visit* | Third visit | Phone call |
|---|---|---|---|---|
| Time points (days after randomisation) | 0 | 5 | At discharge† | 90 |
| Oral and written patient information | + | | | |
| Written consent | + | | | |
| Inclusion/exclusion criteria | + | | | |
| Physical activity (de Morton Mobility Index) | + | + | + | |
| Randomisation | + | | | |
| Demographics | + | | | |
| Installation of accelerometer | + | | | |
| Read-out of accelerometer | | | + | |
| Number of in-hospital falls | | | + / daily | |
| Delirium episodes (EHR) | | | + | |
| Length of stay (EHR) | | | + | |
| Medical history (EHR) | | | + | |
| Destination (EHR) | | | + | + |
| Concerns of falling (FES-I) | + | | | + |
| Independence during activities of daily living (Barthel-Index) | + | | | + |
| Quality of life (EQ-5D) | + | | | + |
| Number of falls (with/without injuries) | | | | + |
| Number of rehospitalisations | | | | + |
| Mortality | | | | + |

*Omitted if third visit before day 6.
†Latest on day 14.
EHR, electronic health record; FES-I, Falls Efficacy Scale-International; SPIRIT, Standard Protocol Items: Recommendations for Interventional Trials.

► Quality of life will be assessed with the EQ-5D-5L questionnaire. The EQ-5D-5L[34 35] is an established, standardised measure of quality of life consisting of a descriptive system comprising the five dimensions mobility, self-care, usual activities, pain/discomfort and anxiety/depression with five levels each (no problems, slight problems, moderate problems, severe problems, and extreme problems) and a visual analogue scale with 0 indicating the worst health and 100 indicating the best health.[39] It can be conducted by telephone interview and a validated German version is available.[36]

► Physical activity will be assessed by a trained study nurse using the DEMMI score. The German version of the DEMMI is a valid[40] and reliable[39] instrument to measure physical activity in older adults in the acute hospital setting and has no floor or ceiling effect.[28 37 41] Based on 15 items (0–19 points with Rasch conversion to a score ranging from 0 (indicating poor physical activity) to 100 (indicating a high level of independent physical activity)), the DEMMI includes evaluation of movement in bed (three items), chair (three), static balance (four), walking (two) and dynamic balance (three). Acute older adult patients with a DEMMI score below 41 score points had significantly lower Barthel index

(38% vs 79%), used more mobility aids, were more bedridden, longer hospitalised and were more often discharged to rehabilitation compared with those with a higher DEMMI score.[42] No association was found for gender, age, weight, body mass index, Charlson comorbidity index or handgrip strength.[42] Trained personnel is able to complete the DEMMI in no more than 10 min.

**Mobility measurements**
To objectively measure mobilisation, patients will be equipped with a wrist-worn tri-axis accelerometer (GENEActiv, Activinsights, Kimbolton, Cambridgeshire, UK). This accelerometer has been validated and proven to reliably measure physical activity in hospitalised patients.[43 44] At inclusion, a study nurse will instruct the patients to wear the accelerometer as long as possible during hospitalisation (day and night time) on either wrist side. The accelerometers will be collected before discharge or transfer to another hospital unit. The measurement frequency will be 50 Hz. Accelerometer data will be extracted and analysed using the GGIR package for R.[45] Time mobilised (total minutes per day) will be calculated from time 'moving' as total of time 'inactive' and 'static', excluding time 'not worn'. Mean acceleration in miliG/vector will be indicated.

## Randomisation and allocation procedures

After patient registration in the data entry system (REDCap) and confirmation of all inclusion and exclusion criteria, patients are randomised 1:1 to one of the two study groups. Randomisation will be blocked using varying block sizes of 2, 4 and 6. Moreover, randomisation will be stratified according to the baseline DEMMI score (≤40 vs >40 points) and age (<65 vs ≥65 years). The allocation sequence (randomisation list) will be generated by an independent data manager at CTU Bern, University of Bern, Bern, Switzerland not otherwise involved in the trial and implemented in the data entry system to ensure concealment of allocation.

## Statistical analysis

The statistical analysis will be performed by CTU Bern, University of Bern, using the statistical software packages Stata or R. All recorded and derived variables will be presented using descriptive summary tables. Continuous variables will be summarised by mean and SD or median and IQR as appropriate. Categorical variables will be summarised with absolute and relative frequencies.

Primary analysis will be intention-to-treat. The primary outcome change in DEMMI score from randomisation to day 5 will be compared between both intervention groups using a linear model. The model will be adjusted for the baseline DEMMI score and stratification factors used for randomisation.

Secondary continuous outcomes will be analysed using the same approach as described above. To assess change in quality of life, we will analyse the difference between baseline and day 5 or day 14 in the visual analogue scale and the mobility dimension of the EQ-5D-5L. Count outcomes (number of falls and rehospitalisations) will be assessed using a negative binomial model, categorical outcomes (destination) using a multinomial logistic model, binary outcomes (reaching MCID) using logistic regression and time-to-event outcomes (LOS, mortality) using a Cox regression model. All models will be adjusted for stratification factors. In a per-protocol analysis, patients that violated any eligibility criteria did not receive the allocated intervention or were discharged before day 4 will be disregarded.

All effect measures will be accompanied by 95% CI. The statistical testing will be two-sided with a type I error of 5%. The data analysts will be blinded to the group.

We will perform subgroup analysis for age groups <65 years vs ≥65 years, initial performance in physical activity (DEMMI≤40 vs DEMMI>40), and prehospital mobility (no mobility aid vs prehospital use of mobility aid) because previous studies showed that older age categories had significantly lower mean DEMMI scores,[46] and lower Barthel index and use of mobility aids or being bedridden was associated with lower DEMMI scores.[42]

Any deviation from the original statistical plan will be described and justified in the final trial report. There is no interim analysis planned, that is, there are no stopping rules on the individual or trial level.

**Table 3** Comparison of JH-HLM and GoMob-in mobility goal levels

| Level | JH-HLM | GoMob-in mobility goal level |
|---|---|---|
| 1 | Only lying | Bed activities |
| 2 | Bed activities | Sit at edge of bed |
| 3 | Sit at edge of bed | Transfer to chair/commode |
| 4 | Transfer to chair/commode | Standing for >1 min |
| 5 | Standing for >1 min | Walking 10+ steps |
| 6 | Walking 10+ steps | Walking 7.5 m or more |
| 7 | Walking 7.5 m or more | Walking 75 m or more |
| 8 | Walking 75 m or more | Walking 75 m or more (30 min or stairs) and no bed rest during daytime |

JH-HLM, Johns Hopkins Highest Level of Mobility Scale.

## Handling of missing data and drop-outs

If outcomes are missing, we will employ multiple imputation in the primary analysis and additionally perform an available case analysis as sensitivity analysis disregarding missing data. The DEMMI score that is measured two times during follow-up will additionally be evaluated in a repeated measures mixed-effects linear model, additionally introducing a random intercept for patients into the model.

## Interventions

*Experimental group*: In patients assigned to GDM, a trained physiotherapist will provide a short educational intervention on GDM. Further, the physiotherapist will define a personal mobility goal based on an adapted version of the Johns Hopkins Highest Level of Mobility (JH-HLM) scale (table 3, online supplemental figure 1).[11 20] To avoid a ceiling effect in our population, compared with the original version of the JH-HLM, we added the following mobility goal: 'Level 8: Walking 75 m or more (30 min or stairs) and no bed rest during daytime' and skipped the original 'Level 1: Only lying'.

Participants will be instructed that they should aim for successful completion of the task at least three times daily. The mobility goal level will be defined regularly by the treating physiotherapist together with the patient taking into account the current physical activity. Reassessment will serve as opportunity for a short motivational intervention ('booster session'). The mobility goal level should be increased regularly. In patients reaching the highest mobility goal level, a note will be placed on the bed to indicate the patient to use the bed only for examinations and for sleep. All medical personnel (eg, nurses) caring for the participant are allowed to adapt the mobility goal level. Increasing the mobility goal level more than one step or lowering is also allowed. The study personnel will instruct nurses and physicians caring for participants in the intervention group regularly about the ongoing trial

and its aims, that is, responsible resident physicians will be informed by email after randomisation of a participant to the intervention group, nurses will be instructed orally and with leaflets at staff meetings at the beginning of the study and if deemed necessary.

The mobility goal level will be depicted on a patient board next to the bed, visible for all stakeholders (patients, visiting friends and family, physicians, nurses, physiotherapists).

*Control group*: Patients assigned to the control group will receive standard treatment of the GIM wards. Mobility goals will not be routinely set nor encouraged interprofessionally. Patients will receive physiotherapist's guided training as prescribed by the resident in charge (resident physician/'Assistenzarzt'). As in any exercise trial, physiotherapists providing the standard treatment cannot be blinded to participation of the patients in the control group. They address the specific needs and provide general non-systematic information about the importance of mobilisation during hospitalisation as part of their routine instruction for all patients.

The adherence to intervention protocol will be assessed by presence of the patient board displaying the mobility goal level at discharge.

## Follow-up

Nurses will record falls in the electronic health record (EHR). Hospital parameters, for example, discharge destination, end of hospitalisation, LOS will be collected from the EHR at discharge. Medical data, for example, number of falls and delirium episodes during hospitalisation, main diagnosis and comorbidities will be systematically collected from the discharge letter.

Physical activity will be assessed using DEMMI in all participants on day 5 after study inclusion and on the day of discharge or day 14 (whichever occurs first).

Three months after study inclusion, all participants will be inquired by telephone interview for Barthel index, FES-I, EQ-5D, number of falls since study inclusion (with/without injuries), current destination, rehospitalisations and all-cause mortality. If the patient cannot be reached, a family member or the treating general practitioner will be called to complement data. The patient will be asked to send the falls calendar to the study team. If discrepancy is noted compared with the phone interview, the participant will be called for validation.

## Blinding

The DEMMI will be assessed by an independent physiotherapist not otherwise involved in the treatment of patients on the ward and to whom the group allocation of the participant is not declared. The patient will be instructed to not reveal the group allocation to the DEMMI assessor.

The statistician responsible for the final analysis will have no access to the group allocation until the primary analysis of the trial is finished. The data analysis will be conducted according to a prespecified statistical analysis plan.

## Data management

Study data will be collected and managed using REDCap electronic data capture tools hosted at CTU, University of Bern, Bern, Switzerland.[47 48] REDCap (Research Electronic Data Capture) is a secure, web-based software platform designed to support data capture for research studies, providing (1) an intuitive interface for validated data capture; (2) audit trails for tracking data manipulation and export procedures; (3) automated export procedures for seamless data downloads to common statistical packages and (4) procedures for data integration and interoperability with external sources.

## Safety

Protocol violations should not lead to treatment discontinuation unless they indicate a significant risk to patient safety. We will comply with all regulations concerning safety measures in clinical trials as set forth by the ethical committee. The investigators will report any serious adverse events occurring during clinical trials, independent of direct causal relationship with the intervention, within 24 hours.

## Data monitoring committee

Data monitoring will be conducted by the CTU Bern, University of Bern, Bern, Switzerland and include routine site monitoring visits according to a specific monitoring plan. Data collection, handling of the data and the analysis will be done by the researchers at the University of Bern, Bern, Switzerland.

**Contributors** FDL designed the study. MMW, AL, MV, JMSL and JH critically commented on the methods and contributed to the development of the study. FDL wrote the first draft of the manuscript. All authors revised the manuscript and approved the final version.

**Funding** The GoMob-in trial is funded by the Swiss Society of General Internal Medicine (SGAIM) Foundation, Bern, Switzerland (grant number: not applicable) and the Fondation Sana, Bern, Switzerland (grant number: GF 2021-0048).

**Competing interests** The funder plays no role in the study design, collection of data, analysis and writing of the manuscript. There is no conflict of interest regarding intellectual content and proprietary affairs. The authors and investigators will have full access to the final trial dataset. No contractual agreements limit such access for investigators.

**Patient and public involvement** Patients and/or the public were not involved in the design, or conduct, or reporting, or dissemination plans of this research.

**Patient consent for publication** Not applicable.

**Ethics approval** The trial will be performed in compliance with the Declaration of Helsinki and its amendments, and it has been approved by the responsible ethics board (Ethikkommission Bern, Bern, Switzerland, 2020-02305). Important protocol modifications will be communicated to the Ethics Board and participating study sites. Potential participants must provide written informed consent before entering the study. Subjects can leave the study at any time for any reason without any negative consequences for their further treatment. When a patient ends the treatment phase of the study prematurely, we will record the date and reason for early treatment discontinuation. If possible, the end of treatment evaluations will be collected. Insurance coverage will be provided for all study participants by the study sponsor. The results of the main trial and each of the secondary outcomes will be submitted for publication in open access peer-reviewed journals. The GoMob-in study group is an independent academic research group, which will not employ

professional writers. Confidentiality of patients' data is ensured by all stakeholders at all times; patients will not be identified by names or identifiers such as birth date in any documents leaving the study site. Whenever data are exported from the database (eg, for analysis), data will be coded.

**Provenance and peer review** Not commissioned; externally peer reviewed.

**Data availability statement** Data are available on reasonable request. All data relevant to the study are included in the article or uploaded as supplementary information. Study protocol, statistical analysis plan, informed consent form will be published. Data will be available on reasonable request.

**ORCID iDs**
Fabian D Liechti http://orcid.org/0000-0003-1006-6903
Maria M Wertli http://orcid.org/0000-0001-6347-0198

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
