## [Reviewer comments · BMJ Open]

ARTICLE DETAILS

TITLE (PROVISIONAL)	Effect of goal-directed mobilization intervention compared with standard care on physical activity among medical inpatients: protocol for the GoMob-in randomized controlled trial
AUTHORS	Liechti, Fabian; Heinzmann, Jeannelle; Schmidt Leuenberger, Joachim; Limacher, Andreas; Wertli, Maria M; Verra, Martin

VERSION 1 – REVIEW

REVIEWER	Gibson, W. University of Alberta Faculty of Medicine and Dentistry, Division of Geriatric Medicine
REVIEW RETURNED	22-Mar-2022

GENERAL COMMENTS	Thank you for the opportunity to review this protocol, describing a non-blinded RCT of early mobilisation in hospitalised patients on Internal Medicine wards in a single centre. Overall the study is well designed and described. I have a few minor comments; The term "geriatric patients" should be avoided - "older adults" is preferred. Geriatric should refer to the unit or physician, not the patient. There is no mention of illness severity as a variable - will this be considered, measured, and included in the analysis? If not this should be explicitly mentioned - for example if the authors feel that randomisation and the sample size is sufficient to avoid an effect. Similarly, how will those receiving interventions which limit mobility but do not require bed rest be handled - supplementary oxygen being the most obvious example? Exclusion criteria - please justify why people with visual impairment are excluded. "Terminal illness" is listed as an exclusion criteria - I assume this means those people admitted for end-of-life care or who are actively in the dying phase, rather than anyone with an incurable illness (such as metastatic cancer). This should be clarified. Will the duration of sleep be considered? There instinctively seems to be a difference in immobility during normal sleep compared to unnecessarily resting in bed while awake. Please clarify the meaning of "resident in charge" - is this a senior nurse? Resident physician in the North American sense (ie, a trainee)? A hospital administrator?
---

REVIEWER	Smart, Denise A. Washington State University - Spokane, College of Nursing
REVIEW RETURNED	27-Mar-2022

GENERAL COMMENTS	Well written protocol study that articulates the intervention and by providing physiotherapy to all patients you have eliminated any potential ethical issues of equity of care. Your intervention is clearly described and goal-setting engages patients in their own care. You have clear measurable primary and secondary outcomes. You added delirium episodes which is very relevant as well as LOS and changes in falls reports. My question has to do with: If sponsored by a Swiss agency, why are you limiting recruitment to German speaking only? Would French also apply? You are recruiting from Switzerland as well (p. 9 Line 16-17). Sample size calculations are appropriate for power and effect. The study is easy to follow, organized well and sequential in steps described. Again later on pages 11-12 you mention German. Perhaps this is a German-speaking dominated hospital or facility? Table 3 is excellent. Page 17: Line 6. I believe you need to change the verb from "is" to "are" since your noun is plural as well (All medical personnel). Page 17: Line 10-11: Can you expand on what "regularly informed" entails? Daily, per shift, 3 times a week?
---

VERSION 1 – AUTHOR RESPONSE

Reviewer: 1

Dr. W. Gibson, University of Alberta Faculty of Medicine and Dentistry Comments to the Author:

Thank you for the opportunity to review this protocol, describing a non-blinded RCT of early mobilisation in hospitalised patients on Internal Medicine wards in a single centre.

Overall the study is well designed and described.

I have a few minor comments;

- The term "geriatric patients" should be avoided - "older adults" is preferred. Geriatric should refer to the unit or physician, not the patient.
 - ▶ Thank you for your comment. We have changed the wording accordingly.

- There is no mention of illness severity as a variable - will this be considered, measured, and included in the analysis? If not this should be explicitly mentioned - for example if the authors feel that randomisation and the sample size is sufficient to avoid an effect.
 - ▶ We plan to record main diagnosis and comorbidities, although we do not plan to use this variable for statistical analysis. We do not plan to use illness severity as a variable because of

the heterogeneity of diseases encountered on a general internal medicine ward. As it is a randomised, controlled trial design with a sufficiently high sample size, we do not think this variable should be added. However, the Barthel index may reflect indirectly severity of disease and will be used as variable.

We have added to the «baseline assessment» section (page 11) the following statement: «Comorbidities will be reported by the participant, using an established, systematic method as previously described. [Sangha O et al. The Self-Administered Comorbidity Questionnaire: a new method to assess comorbidity for clinical and health services research. *Arthritis Rheum* 2003;49(2):156-63. doi: 10.1002/art.10993]»

We have specified in the «Follow-up» section the statement on comorbidities as follows: «Nurses will record falls in the electronic health record (HER). Hospital parameters, e.g., discharge destination, end of hospitalization, LOS will be collected from the EHR at discharge. Medical data, e.g., number of falls and delirium episodes during hospitalisation, main diagnosis, and comorbidities will be systematically collected from the discharge letter.»

- Similarly, how will those receiving interventions which limit mobility but do not require bed rest be handled - supplementary oxygen being the most obvious example?
 - We do not plan to systematically collect data on limitations of mobility, although we are aware, in the acute hospital setting many such limitations will be present, because this is beyond the aim of the present study and it would be difficult to standardise. However, due to the study design used, i.e., randomised, controlled trial, and the number of participants planned to include, this limitation should not relevantly change the validity of the study. Furthermore, this may be part of the intervention itself as barriers to mobilization could be recognized more often in the intervention group and removed, if possible, by the treating physicians/nurses/physiotherapists. We will record mobility by accelerometer to record any potential differences in mobilisation time.
- Exclusion criteria - please justify why people with visual impairment are excluded. "Terminal illness" is listed as an exclusion criteria - I assume this means those people admitted for end-of-life care or who are actively in the dying phase, rather than anyone with an incurable illness (such as metastatic cancer). This should be clarified.
 - Blind patients are excluded because the intervention is mainly based on giving written information on a board besides the patient's bed, i.e., the daily mobility goal. We will not exclude patients with visual impairments who are able to follow study instructions. «Terminal illness» refers to an actively dying phase, as assumed by the reviewer. We include patients with incurable illness. This criterion has been specified in the protocol (page 8) as follows: «Terminal illness (i.e., end-of-life care, dying phase).»
- Will the duration of sleep be considered? There instinctively seems to be a difference in immobility during normal sleep compared to unnecessarily resting in bed while awake.
 - We thank the reviewer for this consideration. The accelerometer data will only distinguish between different states of mobility. We will not be able to record duration of sleep, which would necessitate additional resources / technical equipment we unfortunately do not have. We will consider adding this possible limitation to the discussion of relevant manuscripts.
- Please clarify the meaning of "resident in charge" - is this a senior nurse? Resident physician in the North American sense (ie, a trainee)? A hospital administrator?

► «Resident in charge» is a trained medical doctor, i.e., «Assistenzarzt» / assistant physician / resident physician, who is responsible for the treatment of the patients. We have specified this in the manuscript.

Reviewer: 2

Dr. Denise A. Smart, Washington State University - Spokane Comments to the Author:

Well written protocol study that articulates the intervention and by providing physiotherapy to all patients you have eliminated any potential ethical issues of equity of care. Your intervention is clearly described and goal-setting engages patients in their own care. You have clear measurable primary and secondary outcomes. You added delirium episodes which is very relevant as well as LOS and changes in falls reports.

My question has to do with: If sponsored by a Swiss agency, why are you limiting recruitment to German speaking only? Would French also apply? You are recruiting from Switzerland as well (p. 9 Line 16-17).

► The study site is in the German speaking part of Switzerland and most patients are German speaking. Some potential participants may not be eligible due to lacking German skills, however, as participants will be randomly assigned to one of the two groups, we do not see any confounding problem.

Sample size calculations are appropriate for power and effect.

The study is easy to follow, organized well and sequential in steps described.

Again later on pages 11-12 you mention German. Perhaps this is a German-speaking dominated hospital or facility?

► See above.

Table 3 is excellent.

Page 17: Line 6. I believe you need to change the verb from "is" to "are" since your noun is plural as well (All medical personnel).

► Thank you. We corrected the mistake.

Page 17: Line 10-11: Can you expand on what "regularly informed" entails? Daily, per shift, 3 times a week?

► We have specified this issue in the manuscript as follows: “The study personnel will instruct nurses and physicians caring for participants in the intervention group regularly about the ongoing trial and its aims, i.e., responsible resident physicians will be informed by e-mail after randomization of a participant to the intervention group, nurses will be instructed orally and with leaflets at staff meetings at the beginning of the study and if deemed necessary.

VERSION 2 – REVIEW

REVIEWER	Gibson, W. University of Alberta Faculty of Medicine and Dentistry, Division of Geriatric Medicine
REVIEW RETURNED	05-Apr-2022
GENERAL COMMENTS	Thank you for the opportunity to review this revised protocol. The issues raised by myself and my co-reviewer have been appropriately addressed. I look forward to seeing the results of the trial.